# Multi-level patterning nucleic acid photolithography

Kathrin Hölz [1], Erika Schaudy [1], Jory Lietard [1] & Mark M. Somoza [1]

The versatile and tunable self-assembly properties of nucleic acids and engineered nucleic acid constructs make them invaluable in constructing microscale and nanoscale devices, structures and circuits. Increasing the complexity, functionality and ease of assembly of such constructs, as well as interfacing them to the macroscopic world requires a multifaceted and programmable fabrication approach that combines efficient and spatially resolved nucleic acid synthesis with multiple post-synthetic chemical and enzymatic modifications. Here we demonstrate a multi-level photolithographic patterning approach that starts with large-scale in situ surface synthesis of natural, modified or chimeric nucleic acid molecular structures and is followed by chemical and enzymatic nucleic acid modifications and processing. The resulting high-complexity, micrometer-resolution nucleic acid surface patterns include linear and branched structures, multi-color fluorophore labeling and programmable targeted oligonucleotide immobilization and cleavage.

[1] Institute of Inorganic Chemistry, Faculty of Chemistry, University of Vienna, Althanstrasse 14 (UZA II), 1090 Vienna, Austria. Correspondence and requests for materials should be addressed to J.L. (email: jory.lietard@univie.ac.at) or to M.M.S. (email: mark.somoza@univie.ac.at)

The self-assembly properties of nucleic acids make them important tools for molecular-level fabrication of functional devices[1–3], nanometer-scaled spatial patterning[4,5], circuits[6–8], multiplexed biosensors[9,10], and as scaffolding for functional nanostructures based on organic[11–14] or inorganic building blocks[15,16]. Watson–Crick base-pairing-driven self-organization has the advantage that libraries of even relatively short oligonucleotides can provide large numbers of uniquely addressable sequences. The functionality of oligonucleotides can be further extended by taking advantage of non-Watson–Crick basepairing[17], through chemical derivatization and through conjugation with nanoparticles[18], fluorophores[19] and other biomolecules[20]. While DNA has traditionally been used in bionanotechnology, chimeric sequences incorporating RNA monomers as well as other natural, modified or fully engineered xenobiotic nucleotides can vastly expand the available functionality through modulation of melting temperatures, genetic alphabet expansion, protein recognition and enzymatic processing.

Taking advantage of the full complexity of the chemical and combinatorial space available requires a photolithographic approach that combines flexible and high-resolution, spatially resolved in situ nucleic acid synthesis with additional post-synthetic chemical and enzymatic modifications. Here we report on multi-patterning nucleic acid photolithography, which combines photolithographic synthesis of cleavable or immobile linear or branched nucleic acid constructs, complex crosslinking photolithography, and sequence-addressable enzymatic cleavage and polymerization. Previous work on photolithographic nucleic acid patterning has been limited to glass or silicon surface immobilization of a single pre-synthesized DNA sequence, followed by a photochemical modification step making use of crosslinking agents, photocleavable spacers or the photo-dimerization potential of thymine homopolymers[21–24]. In these approaches, illumination through a photomask results in either a positive or negative tone transfer that is visualized via addition or subtraction of a fluorescent label or results in surface-bound oligonucleotides which can be further functionalized by hybridization with DNA strands or oligonucleotides to generate biologically functional DNA brushes or DNA circuits.

In order to increase the available complexity and functionality of biochip patterning, we introduce multi-level patterning nucleic acid photolithography. This name reflects the combined use of multiple patterning methods that result in a surface complexity and functionality that is orders of magnitude greater than previously possible. The biochips fabricated with this approach now have readily achievable information content densities exceeding 300 megabitscm$^{-2}$, and which can be extended, with available maskless surface patterning photolithography technology, by another three orders of magnitude[25]. The first level of patterning —photolithographic in situ nucleic acid synthesis—is a variant of phosphoramidite chemistry that makes use of monomers with photolabile 5′ (or 3′) hydroxyl protecting groups to replace the standard acid-labile dimethoxytrityl (DMTr) group commonly used in solid phase synthesis. The use of digital micromirrors instead of photomasks allows fast and automated synthesis of hundreds of thousands to millions of unique sequences with a simple desktop photolithographic device with almost no moving parts[26–28]. This synthesis can be performed on almost any flat substrate and is largely independent of the identity of the monomers as long as they share a common coupling chemistry, usually based on the phosphoramidite moiety[29]. This means that DNA nucleosides can be replaced or supplemented by near arbitrary combinations of biogenic and xenobiotic nucleic acid monomers, including RNA, non-canonical DNA and RNA nucleosides, genetic alphabet extensions, mirror-inverted (left-handed) monomers, and nucleic acid analogs engineered with sugar, nucleobase or backbone modifications. In addition, non-nucleic acid monomers, such as fluorescent-, branching-, and reactive-group phosphoramidites can be used to add additional functionality and to generate non-linear polymers (Fig. 1a). The building blocks are assembled into oligonucleotides on the surface through selective removal of o-nitrobenzyl photolabile groups[30] by UV-A light directed with a digital micromirror device (DMD) (Fig. 1b, c). The scale of this first level of patterning is determined

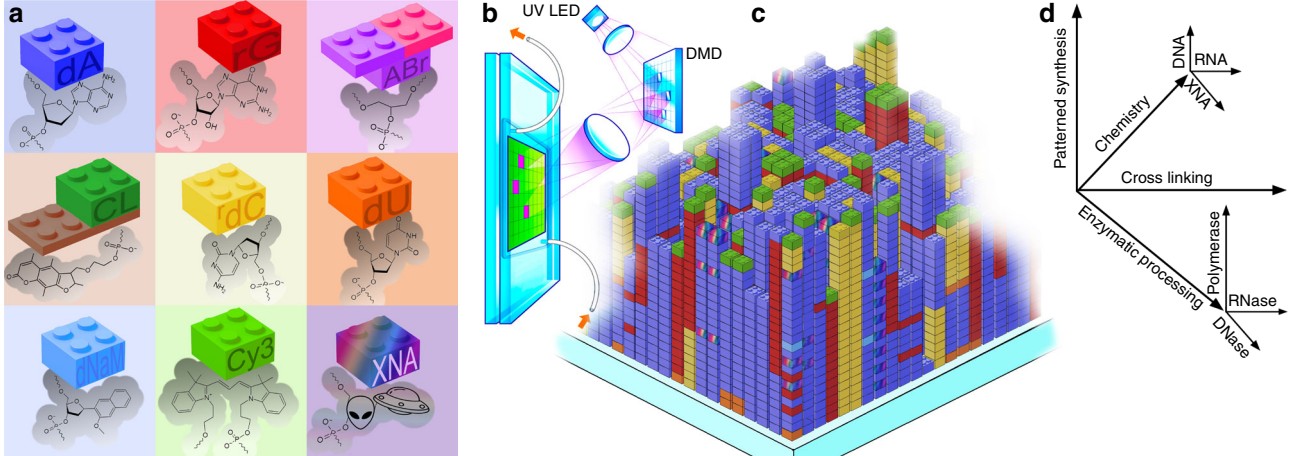

**Fig. 1** Building blocks, photolithographic synthesis and orthogonal processing for multi-patterning nucleic acid photolithography. **a** High-level schematic depiction and molecular detail of representative elements of classes of nucleic acid building blocks available for photolithographic synthesis. These elements include essentially all biogenic nucleic acids (canonical and non-canonical DNA and RNA nucleotides), structural modifiers including branching and spacing monomers, fluorescent and functionalizing modifiers, and engineered xenobiotic nucleic acid (XNA) monomers with modified sugars, bases or backbones. **b** Massively parallel surface synthesis is performed using maskless photolithography mediated by patterned light from a digital micromirror device to selectively remove the photolabile protecting group present on each monomer. **c** Surface synthesis results in dense arrays including hundreds of thousands to millions of unique sequences at defined positions. **d** Multi-level patterning includes multiple orthogonal surface modifications: patterned surface biopolymers synthesis with free choice of monomer chemistry, further modification with additional photochemistry, and final modifications with a variety of nucleic acid processing enzymes ranging from DNA and RNA polymerases to nucleases such as uracil-DNA glycosylase and RNase HII

by the number of micromirrors, which range from $1024 \times 768$ (XGA; 786,432 mirrors) to $4096 \times 2160$ (4 K; 8,847,360 mirrors).

Second-level patterning refers to photochemical modifications after synthesis, particularly photo-crosslinking, as this approach combines self-assembly with the formation of covalent bonds to form complex higher-ordered nucleic acid structures. Multiple crosslinking methods are available, but we focus on the use of psoralen derivatives[31] and the 3-cyanovinylcarbazole ($^{CNV}$K) nucleoside analog[32], as these approaches are known to be effective, have well-understood photochemistry, and are accessible for routine use. Both of these approaches require specific nucleoside contexts, opposing TA base pairs in the case of psoralen and an opposing pyrimidine in the case of $^{CNV}$K, but this is not a significant limitation since our approach provides full control of sequence space. We will show that both approaches provide very similar yields even though the crosslinking quantum efficiency of $^{CNV}$K is far higher than that of psoralen. The third level of patterning involves sequence modifications performed by nucleic acid processing enzymes. Such processing includes the use of DNA and RNA polymerases and ligases to selectively add new nucleic acid biopolymers to the biochip and DNA and RNA exo- and endonucleases to selectively remove biopolymer sections from the surface (Fig. 1d). As an example of such enzymatic processing, we demonstrate the use of a DNA polymerase to extend primers attached to template strands synthesized in situ on the biochip surface.

## Results

**Level-One patterning nucleic acid photolithography.** To demonstrate the very high level of multiplexing versatility of Level-One patterning, we used in situ DNA synthesis to reproduce, at high resolution and at a highly miniaturized scale, a complex color image. To generate a color image, we separated the original image into red, green and blue channels and assigned each gray level within each channel to DNA sequences fully or partially complementary to Cy5 (red), Cy3 (green) or fluorescein (blue) labeled oligonucleotides. For simplicity, only 8 shades of gray (intensity levels) per channel were used, resulting in a red-green-blue color model (RGB) reproduction with 512 unique colors, each with one of 512 surface-bound oligonucleotides (Fig. 2). To generate the grayscale, we chose 8 oligonucleotides per channel with different melting temperatures that were calibrated to result in 8 uniformly spaced fluorescence intensities when hybridized with the common labeled complement (Supplementary Table 1). We used sequence truncations rather than mismatches to reduce melting temperatures, as this approach is simpler and reduces biochip synthesis time. The depth of the color space can be increased beyond 512 by also using mismatches to generate finer gradations of melting temperatures. Accurate color rendition requires a calibration curve generated experimentally on a biochip since fluorescence intensity also depends strongly on the specific nucleobase context (Supplementary Figure 1)[33]. This first level of photolithographic patterning is insensitive to the number, complexity and layout of oligonucleotides. Oligonucleotide length is essentially unrestricted, although synthesis time is approximately proportional to sequence length. In the case of the image in Fig. 2 (up to 85mers), the synthesis time was ~3 h.

Use of additional nucleic acid building blocks (beyond the four canonical DNA nucleotides) is generally unproblematic but can increase synthesis time[17,34]. Some building blocks, particularly RNA monomers, but also fluorescent, branching and spacer monomers can require coupling times in a range between two and five minutes (vs. 15 s for DNA)[34,35]. This increase can be attributed to steric hindrance at the coupling site by the 2′ hydroxyl protecting group in the case of RNA, overall bulkiness of the monomer (particularly for fluorescent groups) and monomer impurity due to age or manufacturing complexities. In addition, synthesis time increases with the use of additional monomers due to the need for additional coupling and photo-deprotection cycles. Examples of photolithographic synthesis using extended sets of building blocks are introduced below.

**Level-Two patterning nucleic acid photolithography.** Level-One patterning is the dominant source of complexity in nucleic acid photolithography, but post-synthetic photochemical processing, or second-level patterning, can be very useful in creating higher-order functional constructs. Level-Two crosslinking reactions can use the same UV light source but do not interfere with Level-One patterning since crosslinking proceeds only after synthesis on the biochip is complete and no photoactive groups remain on the surface. Furthermore, we can take advantage of the versatility of the maskless photolithographic system used for Level-One patterning to increase the complexity of Level-Two patterning. In particular, positioning accuracy (registration) of the substrate in the imaging system is sufficient to correctly overlay second-level patterning after hybridization with crosslinkable oligonucleotides. In addition, pixel-level UV exposure control is precise and useful for creating complex light exposure gradients and hence for introducing complex crosslinking patterns with grayscale. Thus, two dimensions of crosslinking control are available: through sequence (multiple orthogonal and crosslinkable sequences

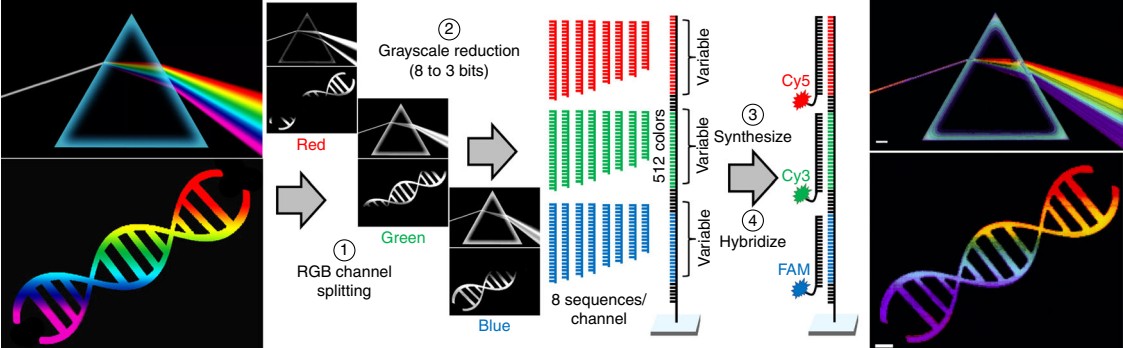

**Fig. 2** Large-scale nucleic acid photolithography (Level-One patterning). Rendering in DNA of a color image. Original RGB is split into individual color channels, each of which is reduced to 8 shades of gray. Each shade of each color channel is assigned a DNA sequence with a calibrated melting temperature with a fluorescently-labeled complementary oligonucleotide. A sequence for each pixel is constructed with one of the resulting 512 variants. After photolithographic synthesis of all the DNA sequences, the surface is hybridized with the three complementary or partially complementary Cy3-, Cy5- or fluorescein-labeled oligonucleotides. The scale bars in the resulting fluorescence images are 200 μm

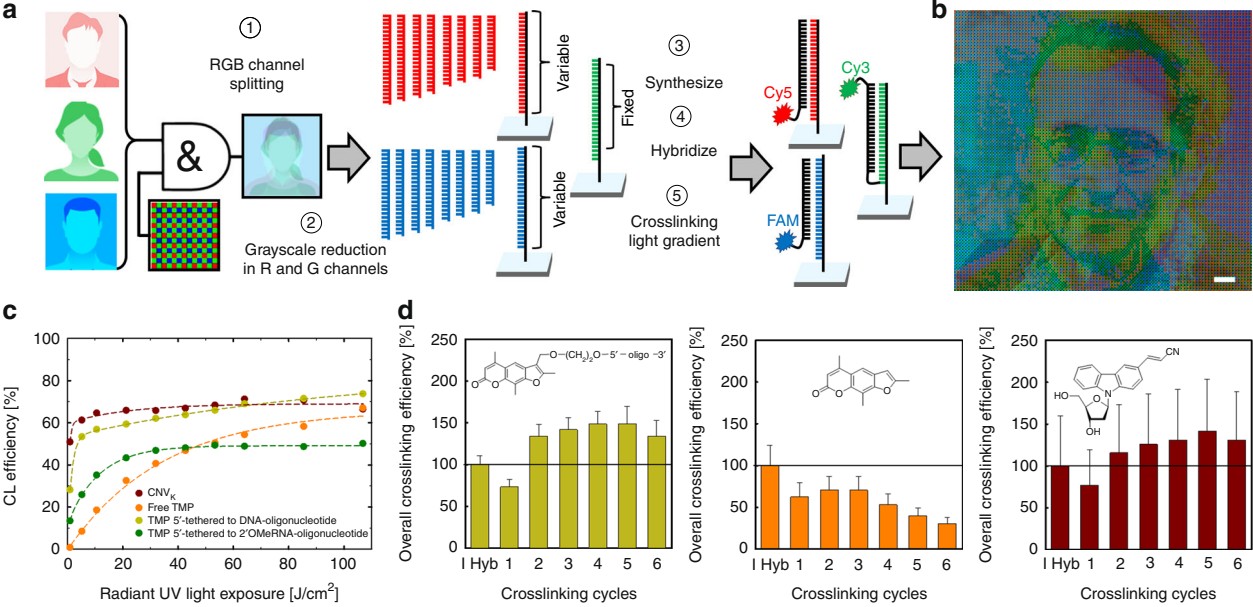

**Fig. 3** Post-synthetic photochemical processing (Level-Two patterning). High complexity patterns can be superimposed after photolithographic synthesis, e.g., via hybridization and selective crosslinking. **a** Nucleic acid crosslinking steganography. Alternatively to oligonucleotides encoding all colors, color-wise ANDing with Bayer filter pattern separates each channel into independent pixels. Two channels (R & B) are encoded using a melting-temperature grayscale and the green channel is encoded using a single sequence followed by a light gradient to impose a grayscale through variable crosslinking efficiency. **b** "Hidden figure" image of three Viennese physicists (200 μm scale bar) is revealed with a green filter or by washing off the non-crosslinked red and blue channels (Supplementary Figure 7). **c** 365 nm UV light exposure gradients for $^{CNV}$K-modified oligonucleotides (dark red), free psoralen (orange), psoralen-modified DNA oligonucleotides (light green) or psoralen-modified 2′-OMeRNA oligonucleotides (dark green). **d** Crosslinking efficiency for tethered and $^{CNV}$K-modified oligonucleotides can be greatly increased with multiple hybridization/exposure cycles; percent values are relative to the first hybridization. Error bars correspond to the standard deviation. Source data are provided as a Source Data file

hybridize at defined surface positions) and through UV control (selective crosslinking efficiency).

This additional level of patterning control is illustrated in Fig. 3 with an example of nucleic acid crosslinking steganography, in which an image is concealed within another image. In this case, we encoded three grayscale images separately into red, green and blue channels. The red and blue channels were encoded similarly to those in Fig. 2, using 8 surface-synthesized oligonucleotides per channel (3-bit color). A single DNA sequence was used for the green channel, with the grayscale obtained by hybridizing and crosslinking a Cy3-labeled oligonucleotide using pixel-specific irradiance control (Supplementary Table 3). As an alternative to encoding all colors into long oligonucleotides on each pixel, this steganograph's channels were generated analogously to the Bayer array filtering used in digital imaging; each of the three grayscale portraits was logically ANDed with a Bayer filter pattern to create a single image with each portrait separated into independent pixels and color channels. After synthesizing the corresponding biochip, the red and green images were reconstructed via hybridization with Cy5- and fluorescein-labeled oligonucleotides. The hidden image, in the green channel, was permanently attached to the chip by hybridization and crosslinking (with grayscale) to a Cy3-labeled oligonucleotide. The three portraits are of well-known Viennese physicists. The hidden image is revealed by washing away the two non-covalently bound colors (Supplementary Figure 7).

Several crosslinking chemistries are appropriate in this context (Supplementary Figure 2). Among these, psoralen C2, with an ethylene linker to the phosphoramidite moiety, is probably the most ubiquitous. Since it is tethered to an oligonucleotide, the position of the crosslink is well defined but requires a near opposing thymine, and preferentially 5′-TA. An alternative is untethered psoralen,

which potentially crosslinks any opposing thymines upon UV exposure; this approach has the advantage of requiring only unmodified nucleic acids and introducing multiple crosslinks per hybrid. 3-cyanovinylcarbazole ($^{CNV}$K) nucleosides[32] were recently introduced as a highly photoreactive alternative to psoralen and have the additional advantage of crosslinking with opposing pyrimidines of either type. As crosslinking elements for second-level patterning, we investigated these three options for crosslinking DNA/DNA hybrids. Additionally, we used psoralen-modified oligonucleotides for photo-crosslinking of DNA/2′-O-methyl RNA hybrids. The use of 2′-O-methyl RNA (2′-OMeRNA) was of particular interest to us in the context of Level-Three patterning, in which we can use DNases to selectively digest DNA while retaining crosslinked 2′-OMeRNA (vide infra). The sequences of complementary oligonucleotide probes used for crosslinking are given in Supplementary Table 2. As shown in Fig. 3c, d, all three crosslinking approaches reach ~70% crosslinking on the chip surface, but $^{CNV}$K reaches this values much faster than tethered psoralen, which in turn is much faster than free psoralen. We hypothesized that the crosslinking yield could be improved by repeated attempts, and indeed, this is the case for tethered psoralen and $^{CNV}$K, but not for free psoralen (Fig. 3d) (see also Supplementary Figures 3–6). Since the biochip oligonucleotides are stably bound to the surface, it is straightforward to wash off non-crosslinked complementary strands, re-hybridize and re-expose. The extent, or yield, of crosslinking of both tethered psoralen and $^{CNV}$K can be doubled—to about 150% of hybridized duplex values—after a few cycles. This value likely corresponds to full crosslinking of all hybridizable oligonucleotides. Although both of these agents are highly effective, we chose tethered psoralen for creating the crosslinking image as its reduced light sensitivity allows for the creation of more accurate crosslinking grayscales.

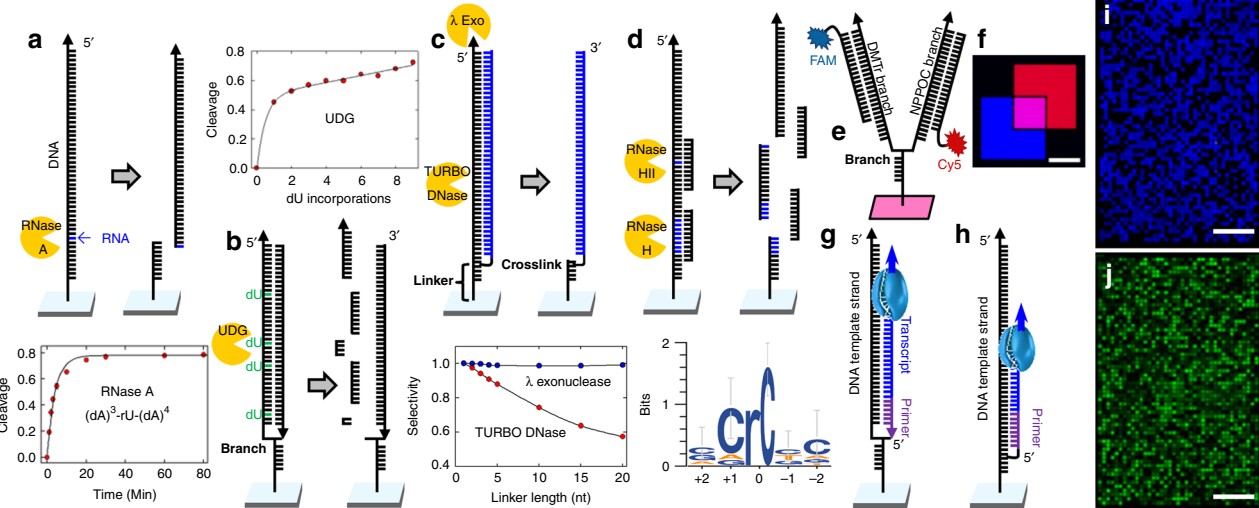

**Fig. 4** Post-synthetic enzymatic processing (Level-Three patterning). Nucleic acid sequences can be selectively or programmably cleaved, digested, or transcribed enzymatically on the biochip. **a** RNase A is a fast and efficient cleavage agent and requires only a single RNA nucleotide incorporation, optimally a pyrimidine surrounded by purines. **b** Similarly, uracil-DNA glycosylase (UDG) cleaves at pre-defined dU positions, but is significantly less efficient than RNase A cleavage. **c** Unmodified DNA can be digested by DNase I or lambda exonuclease. In the case of branched or crosslinked nucleic acid structures, lambda exonuclease digests only from a 5′ terminus and does not pass the junction, leaving the 3′ terminal DNA branch intact. DNase can be used if the second branch is not DNA and will not significantly digest a DNA surface linker shorter than a few nucleotides. **d** Programmable logical control of cleavage can be achieved with RNase H or HII; cleavage only occurs in conjunction with DNA hybridization to the cleavage site. RNase HII requires only a single RNA nucleotide incorporation at the desired cleavage site but has a preferred sequence context for maximum efficiency. **e** Branched structures synthesized either with asymmetric branching phosphoramidites or through crosslinking, can be effectively combined with enzymatic processing. **f** Fluorescence scans showing branched structure labeled by hybridization to either or both branches (200 μm scale bar). **g** Transcription of a DNA template using a branched structure. The primer is synthesized on the second branch using reverse phosphoramidites, or **h** attached by crosslinking. In either case the primer can be elongated as DNA or RNA using appropriate polymerases. **i** Fluorescence scan of branched structured synthesized *via* templated primer elongation using a DNA polymerase and fluorescein labeled NTPs. **j** Fluorescence scan of DNA strand synthesized enzymatically and hybridized after template degradation with UDG (200 μm scale bars). Source data are provided as a Source Data file

**Level-Three patterning nucleic acid photolithography**. The complexity of biomolecular surface patterning and applications can be greatly extended through enzymatic processing. Level-Three patterning consists of enzymatic modifications of nucleic acid structures achieved with the first two levels. The two primary modifiers are nucleases and polymerases.

Figure 4 illustrates the scope of the available modifications. Enzymatic cleavage can be pre-programmable (RNase A, uracil-DNA glycosylase), non-specific (DNase, lambda exonuclease), or post-programmable (RNase H/HII). RNase A is among the most robust and efficient nucleases, requiring only a single RNA nucleotide in a DNA context (Fig. 4a, Supplementary Table 4). Cleavage is fastest with the preferred substrate of rU in a dA context[36]. Uracil-DNA glycosylase (UDG) requires the more straightforward incorporation of a deoxyuridine during synthesis, but the glycosidic bond cleavage is inefficient and must be followed by the endonuclease VIII-mediated cleavage of the phosphodiester bond at the abasic site (Fig. 4b). Nevertheless, it is particularly useful for site-specific cleavage of DNA in biochip DNA-RNA hybrid systems, such as those resulting from on-chip transcription. Non-specific digestion of DNA on biochips can be accomplished with DNase I or its highly efficient engineered variant TURBO DNase, and lambda exonuclease. All such enzymes combine well with branched structures assembled via crosslinking since lambda exonuclease cleaves from the 5′ end of the template strand and is stopped by the junction, preserving the attachment to the surface of the second strand even when this is also DNA (Fig. 4c). Turbo DNase can be used for template degradation when the second strand is non-DNA. It has the potential to also cleave any DNA linkers to the surface, but this

can be mitigated by keeping the linker length between the branch/crosslink and the surface as short as possible (Supplementary Figures 8, 9). Much more versatile are RNase H and HII, both of which allow post-programmable cleavage, i.e., the desired sequence can be targeted for cleavage after synthesis by the addition of the complementary DNA sequence along with the enzyme. RNase HII has the advantage of requiring just a single RNA nucleotide incorporation but is most efficient in specific base contexts as shown in the consensus sequence (Fig. 4d)[34]. On-chip enzymatic synthesis from templates and primers synthesized on or added to the surface is a gateway to the preparation of double-stranded constructs, with both strands attached to the surface, but can also be a useful alternative for the synthesis of 5′→3′-oriented single-stranded nucleic acids that are otherwise difficult or inefficient to synthesize chemically[37,38]. Surface attachment of the enzymatically synthesized strands is accomplished through primer crosslinking or in situ synthesis with NPPOC/DMTr asymmetric branching phosphoramidites (Fig. 4e), with 3′ to 5′ template synthesis on the NPPOC branch and 5′ to 3′ primer (reverse) synthesis on the DMTr branch. Branch synthesis is efficient and versatile, allowing full control of the sequence space on both sides. Figure 4f shows a fluorescent image with oligonucleotide synthesis on either or both branches. Enzymatic synthesis with primers on a branch or crosslinked is illustrated in Fig. 4g, h. In addition to DNA, both enzymatic RNA and 2′-OMeRNA synthesis can proceed by primer elongation with both natural and engineered polymerases (Supplementary Figures 10–11, Supplementary Table 5)[38,39]. On-chip biosynthesis of DNA from a DNA template and branched primer is shown as fluorescence from incorporated fluorescein-labeled NTPs (Fig. 4i)

and, after UDG-mediated template degradation, hybridization of the new strands with Cy3-labeled complementary oligonucleotides (Fig. 4f).

## Discussion

We have introduced a three-level approach to synthesize and manipulate nucleic acid biochips of unparalleled complexity and versatility. The first level of patterning involves the in situ synthesis of complex patterns of nucleic acid sequences onto a planar surface using photolithographic approaches borrowed from the integrated semiconductor industry. Sharing a common phosphoramidite chemistry, as well as photolabile protecting groups, nearly any combination of nucleic acid monomers and non-nucleosidic monomers can be efficiently combined on a common surface or within a sequence on this surface. Traditionally, nanobiotechnology has relied almost exclusively on DNA, but inclusion of other building blocks in the synthesis toolbox results in greatly expanded functionality with modest additional synthesis complexity due to the shared coupling chemistry. The additional functionality originates from greater base-pairing options, such as that introduced by alphabet extensions (dNaM/d5SICS[40] or Hachimoji DNA[41]), from manipulation of duplex stability through nucleic acid sugar modifications (LNA[42] or UNA[43]), from RNA[44] and other natural but non-canonical nucleic acids, from reverse synthesis ($5' \rightarrow 3'$)[45] and structural modifiers including spacers and symmetric or asymmetric branching phosphoramidites, and from other engineered constructs with heterogeneous functionalities, such as mirror-image nucleic acids[46], fluorescent phosphoramidites[47], and binary-encoding phosphoramidites[48]. The use of photolithography in this first level of patterning allows near arbitrary synthesis flexibility and monomer choice, with only near-UV-absorbing modifications, such as pyrene and phenanthrene[49] possibly disallowed due to absorption spectrum overlap with the photolabile protecting groups. The specific photolithographic approach we used, and which we consider most appropriate for nucleic acid biochip synthesis, is termed maskless array synthesis (MAS). MAS makes use of digitally controlled micromirrors, within an optical imaging system, to direct light to the synthesis surface. These systems are far simpler than mask aligner or stepper systems and are suitable for laboratory use. Our current setup is limited to a pixel size of 14 μm, but related maskless photolithographic systems are available with sub-micron pixel size[25,50].

The second level of patterning is the post-synthetic photochemical modification of the biochip, primarily using crosslinking reactions to add covalently attached branches to the surface. Here, complexity is added, first by adding one or many sequences that then hybridize to the desired targets. A crosslinking agent enables the formation of a covalent bond upon UV exposure. Crosslinking complexity is increased by selective exposure to UV light using the same photolithographic system used for the initial patterning. Positioning accuracy is sufficient for correct overlay of synthesis pixels and crosslinking pixels. Furthermore, the pixel-level UV light exposure can be controlled to enable light exposure patterns and hence for introducing complex crosslinking reactions with grayscale, i.e., the degree of crosslinking can be controlled for each pixel. We experiment with three crosslinking chemistries: psoralen C2 phosphoramidites attached to the 5′ end of the oligonucleotide, free psoralen added to the surface during hybridization, and 3-cyanovinylcarbazole ($^{CNV}$K) introduced into the oligonucleotide via the corresponding phosphoramidite. We found that free psoralen is a relatively ineffective crosslinking agent while both psoralen C2 and $^{CNV}$K work very well and can, with repeated cycles of hybridization and crosslinking, reach duplex populations of about 150% relative to uncrosslinked hybridization. This can be attributed to kinetics of duplex

formation in which the back reaction rate is severely reduced by the stable linkage. As a demonstration of the degree of crosslinking control in this approach, we created a nucleic acid crosslinking steganography pattern that superimposes three portraits in the three color channels. Two portraits are created via hybridization only and the third is permanently fixed using a crosslinking grayscale. This last image is obscured by the first two but can be revealed by washing away the oligonucleotides bound by hybridization alone. Crosslinking steganography is an interesting but insecure concealment approach and far more robust molecular encryption approached using nucleic acid biochips are possible[51]. The main applications we envision are for creating complex arrays of thermally stable branching nucleic acid structures that can be used as functional devices such as logic gates, or as stable scaffolding for organic or inorganic building blocks.

The third level of patterning makes use of enzymes, particularly RNA and DNA nucleases and polymerases, to modify the nucleic acid patterning accomplished with the first two levels. Other enzymes, including ligases and translation systems, are possible, but were not explored in this work. Nucleases are useful for removing templates or messenger strands after synthesis of complementary DNA or RNA strands, or after translation with gene expression systems. This patterning would be an integral component of complex, biochip-based spatially organized genetic circuits. In addition, cascading nucleic acid computing circuits based on programmable and targeted substrate cleavage can be achieved with selective nucleases such as RNase H and HII, which can be triggered to cleave at specific locations and release specific sequences for triggering, e.g., downstream logic gates[8,52].

In summary, leveraging high-resolution surface pattern generation methods developed and continuously advanced by the integrated semiconductor fabrication industry, along with an efficient and robust version of phosphoramidite chemistry with a diverse set of natural and engineered monomers bearing photolabile protecting groups it is possible to create complex and versatile nucleic acid biochips with near complete control of chemistry, sequence and structure. These arrays can then be further processed using crosslinking chemistries and enzymatic reactions to efficiently create functional biochips with micron-scale resolution. Potential applications include biosensing devices, complex and spatially organized cell-free gene expression systems, and integrated cascaded nucleic acid circuits for biological computation.

## Methods

**Photolithographic synthesis of nucleic acid biochips.** The custom-built biochip synthesizer uses an Offner optical relay to image the exposure pattern displayed on a Texas Instruments XGA-resolution digital micromirror device (DMD) onto the synthesis surface. The surface is an N-(3-triethoxysilylpropyl)-4-hydroxybutyramide-functionalized glass microscope slide that also serves as the light-entrance window into the reaction chamber where the chemistry and photochemistry take place. The DMD is illuminated with spatially homogenized light from a 365 nm LED (Nichia NVSU333A). Photodeprotection was carried out using 6 J/cm² or 3 J/cm² for NPPOC- or Bz-NPPOC-protected phosphoramidites, respectively. An Expedite 8909 nucleic acid synthesizer pumps reagents and solvents in synchrony with the UV light exposures from a 365 nm LED[53]. After synthesis, the DNA protecting groups are removed with a 2 h immersion in 1:1 (v/v) ethylenediamine (EDA)/ethanol (EtOH) solution. RNA-containing biochips require an additional two-step deprotection preceding EDA/EtOH: (1) agitation in 2:3 (v/v) anhydrous triethylamine in acetonitrile for 90 min., (2) immersion in 0.5 M hydrazine hydrate in 3:2 (v/v) pyridine/acetic acid for 2 h. Phosphoramidites with photolabile protecting groups were sourced from Orgentis Chemicals (DNA monomers) and ChemGenes (RNA, reverse DNA, dU and branching monomers). All other reagents and solvent for the modified phosphoramidite chemistry were obtained from Biosolve and Sigma-Aldrich. Sequences and detailed methods are provided as Supplementary Methods.

**Hybridization and photochemical crosslinking.** Cy3-, Cy5-, FAM-, psoralen or $^{CNV}$K-labeled DNA and 2′-OMeRNA modified oligonucleotides for hybridization and crosslinking were manufactured and HPLC purified by Eurogentec. Hybridization in self-adhesive chambers (Grace Bio-Labs) were performed using standard

buffered solutions for 2 h at 42 °C[54] for fluorescent oligonucleotides, for 30 min for psoralen or $^{CNV}$K-labeled DNA, and for 60 min for crosslinkable 2′-OMeRNA. Scanning was performed at 2.5 μm resolution using appropriate lasers and filters for red, green and blue fluorescence imaging. In the case of crosslinking oligonucleotides, hybridization was followed by exposure to spatially homogenized 365 nm light. For untethered psoralen, a 12.5 μM solution in MES buffer was introduced to the surface of the pre-hybridized biochip. After crosslinking, the surfaces were stringently washed to remove non-crosslinked oligonucleotides. In the case of repeated crosslinking cycles, the surfaces were stringently washed and then rehybridized prior to UV exposure in each cycle.

**Enzymatic strand cleavage**. Lambda exonuclease, RNase H, RNase HII and uracil-DNA glycosylase were purchased from New England Biolabs and TURBO DNase from Invitrogen and were used at the recommended concentrations and with supplied buffers. RNase A was obtained from Sigma-Aldrich and applied at 100 nM concentration in 0.1 M MES buffer. For each biochip cleavage assay, appropriate nucleic acid substrates were synthesized. In the case of double-stranded substrates, the biochip was synthesized with oligonucleotides capable of self-annealing to form hairpin-loop structures. Cleavage was detected through either the loss of fluorescence of Cy3-terminally labeled sequences, or loss of fluorescence from Cy3-labeled oligonucleotides hybridized to the nuclease substrate. In time course experiments, the enzymatic reaction was stopped by washing the surface followed by drying with argon.

**In situ polymerization**. The template DNA strands were synthesized photolithographically. The primer was either synthesized in situ using a NPPOC/DMTr asymmetric branching phosphoramidite (ChemGenes) with the template on the NPPOC branch and the primer on the DMTr branch, or by hybridizing and crosslinking the primer to the template strand. For the former, the template strand was synthesized first and then capped with a diethyleneglycol ethyl ether phosphoramidite (Glen Research); after a standard deblocking step with dichloroacetic acid in dichloromethane for 30 s, the primer was added to the second branch using reverse phosphoramidites. For enzymatic DNA synthesis, we used DNA Polymerase I, large (Klenow) fragment, or the Therminator DNA Polymerase along with a primer extension mix. RNA polymerization requires the use of T7 RNA Polymerase, ribonucleoside triphosphates (NTPs), and 2′-OMeRNA or RNA primers[38]. All polymerases, triphosphates and polymerization buffers were obtained from New England Biolabs.

## Data availability
Data for the figures in this manuscript can be found in the Source Data file. The data that support the findings of this study are available from the corresponding authors on reasonable request.

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

## Acknowledgements

Funding by the University of Vienna, the Faculty of Chemistry of the University of Vienna, the Austrian Science Fund (grant FWF P23797, P27275 and P30596) is gratefully acknowledged.

## Author contributions

M.S. and J.L. conceived the experiments. K.H., E.S. and J.L. performed the experiments concerning biochip synthesis and crosslinking. K.H. and J.L. performed the experiments on enzymatic processing. All authors undertook data analysis and discussed the results. K.H. and J.L. contributed to drafts of the manuscript. M.S. wrote the final manuscript.

## Additional information

**Competing interests:** The authors declare no competing interests.

