## [Peer Review File · Nature Communications]

Reviewers' Comments:

Reviewer #1:

Remarks to the Author:

Holz et al reported a multi-level photolithographic patterning approach to develop LED-like DNA display, which results in high-complexity, micrometer-resolution nucleic acid surface patterns. This work presents one more excellent example for showing the power of programmable DNA reactions. I generally support the publication of this work in Nature Communications, however I have the following concerns that need to be addressed:

1. At first the author introduced a novel DNA synthesizing protocol which they described as "photolithography". That is very interesting. However, I see little information about the protocol. What's the maximum length of oligonucleotides synthesized in this protocol? What is the synthesizing efficiency? Is there interference between different positions during DNA synthesizing?
2. The authors claimed that "Multiple crosslinking methods are available, but we focus on the use of psoralen derivatives and the 3-cyanovinylcarbazole (CNVK) nucleoside analog" on line 93 to 94 in page 4. They should present the reasons that they chose them.
3. The authors claimed that "We will show that both approaches provide very similar yields even though the crosslinking efficiency of CNVK is far higher than that of psoralen" on line 96 to 97 in page 4. What are the yields?
4. I am confused about the reconstruction of the painting in figure 2. I understand the color in individual pixel is dependent on the RGB composition which is introduced by three fluorophore-labeled oligonucleotides. But what's the relationship between the color and the melting temperatures of the oligos? Did the authors use temperature control?
5. In level-two painting, the authors further introduced UV exposure to induce crosslinking of oligos. But the UV exposure has been introduced for activating DNA synthesis in level-one painting. Is there any interference about the repeated use of UV light?
6. Why does the crosslinking efficiency decrease as cycles accumulated in figure 3d?
7. The author claimed "and indeed, this is the case for tethered psoralen and CNVK, but not for free psoralen (Fig. 3d)" on line 158 to 159 in page 6. However, figure 3d doesn't give any evidence on it. I am also confused why the figure 3c is identical to figure S3.
8. Finally, some subfigures (e. g., figure 3a, b) were not described in the main text, and many figures in the SI were not quoted in the main text.

Reviewer #2:

Remarks to the Author:

Authors developed a three-level photolithographic patterning approach: first, photolithographic in situ nucleic acid synthesis for maskless fabrication of oligonucleotide microarrays by a digital micromirror array technique, second photo-crosslinking, and third sequence modifications by nucleic acid processing enzymes. A potential 512-color palette on a grid of 256 × 332 pixels was realized for a Picasso's *Femme assise* painting in the first level, while nucleic acid crosslinking is used for steganography to conceal an image in a complex pattern in the second level. The results shown in the manuscript are of broad interest to the nano-device community for creation of functional biochip by controlling chemical identity, nucleic acid sequence and surface patterns. I recommend publication of the manuscript after substantial revisions.

1. My major concern is that in the first level, although a painting of Picasso's *Femme assise* is marvelous by a DNA hybridization approach, author should not offer this as an example for validity of their method. Instead, they should give a details for the principle of their palettes. How in principle to realize the different colors? and how in experiment to realize the component or ratio of RED, GREEN and BLUE? They should compare the colors they obtained to the standard colors by selected colors, even though not totally 512 colors.
2. 42 °C was chosen for hybridization temperature in manufacturing microarray, why? What happen if at other temperatures?
3. In the third level, authors should provide an example for application, as that of image

steganography in second level.

Reviewers' comments:

Reviewer #1 (Remarks to the Author): Holz et al reported a multi-level photolithographic patterning approach to develop LED-like DNA display, which results in high-complexity, micrometer-resolution nucleic acid surface patterns. This work presents one more excellent example for showing the power of programmable DNA reactions. I generally support the publication of this work in Nature Communications, however I have the following concerns that need to be addressed:

Authors comment: We thank the reviewer for carefully and critically reading our manuscript and for recommending its publication in Nature Communications.

1. At first the author introduced a novel DNA synthesizing protocol which they described as “photolithography”. That is very interesting. However, I see little information about the protocol. What’s the maximum length of oligonucleotides synthesized in this protocol? What is the synthesizing efficiency? Is there interference between different positions during DNA synthesizing?

Authors reply: The maximum length used in these experiments was 85mers. This was mentioned right before Figure 2. Nevertheless length and efficiency as well as cross-talk between adjacent positions are important considerations in any synthesis approach, so we have added a section in the supplementary information (SI 1. Synthesis efficiency and yield) with additional details. This is a complex topic and we are still working to fill in gaps in our knowledge of synthesis error rates as a function of various experimental parameters by sequencing oligonucleotides synthesized using this photolithographic approach; we hope to publish these results in the next year.

2. The authors claimed that “Multiple crosslinking methods are available, but we focus on the use of psoralen derivatives and the 3-cyanovinylcarbazole (CNVK) nucleoside analog” on line 93 to 94 in page 4. They should present the reasons that they chose them.

Authors reply: We chose these because they are commercially available, known to be effective, and have known photochemistry. To clarify this point in the manuscript we have made the following change to the relevant sentence: “Multiple crosslinking methods are available, but we focus on the use of psoralen derivatives³⁰ and the 3-cyanovinylcarbazole (CNVK) nucleoside analog³¹, as these approaches are known to be effective, have well-understood photochemistry, and are accessible for routine use.”

3. The authors claimed that “We will show that both approaches provide very similar yields even though the crosslinking efficiency of CNVK is far higher than that of psoralen” on line 96 to 97 in page 4. What are the yields?

Authors reply: This sentence is just a preview of the results described on page 6, shown in Figure 3 (and in greater detail in the SI, Section 2). But the reviewer is probably being confused by our misleading use of the word “efficiency” when we mean “yield” in the figures. To address this, we have made the following change to the text on page 6: “The extent, or yield, of crosslinking of both

tethered psoralen and CNVK can be doubled—to about 150% of hybridized duplex values—after a few cycles.” And we have also changed “efficiency” to “yield” on the figure vertical axes.

4. I am confused about the reconstruction of the painting in figure 2. I understand the color in individual pixel is dependent on the RGB composition which is introduced by three fluorophore-labeled oligonucleotides. But what’s the relationship between the color and the melting temperatures of the oligos? Did the authors use temperature control?

Authors reply: Yes, as described above Figure 2 and in the Figure 2 caption, the grayscale within each of the colors corresponds to varying melting temperatures of the oligonucleotides. Table S1 in the supplementary information gives the sequence responsible for each of the 8 levels of grayscale per color channel. To clarify this point, we have elaborated our description on page 4:

“To demonstrate the very high level of multiplexing versatility of Level-One patterning, we used in situ DNA synthesis to reproduce, at high resolution and at a highly miniaturized scale, a well-known painting by Picasso. To generate a color image, we separated the original image into red, green and blue channels and assigned each grey level within each channel to DNA sequences fully or partially complementary to Cy5 (red), Cy3 (green) or fluorescein (blue) labeled oligonucleotides. For simplicity, only 8 shades of grey (intensity levels) per channel were used, resulting in a red-green-blue color model (RGB) reproduction with 512 unique colors encoded on a grid of 256 × 332 pixels, each with one of 512 surface-bound oligonucleotides (Fig. 2). To generate the greyscale, we chose 8 oligonucleotides per channel with different melting temperatures that were calibrated to result in 8 uniformly spaced fluorescence intensities when hybridized with the common labeled complement. We used sequence truncations rather than mismatches to reduce melting temperatures, as this approach is simpler and reduces biochip synthesis time. The depth of the color space can be increased beyond 512 by also using mismatches to generate finer gradations of melting temperatures. Accurate color rendition requires a calibration curve generated experimentally on a biochip since fluorescence intensity also depends strongly on the specific nucleobase context³². This first level of photolithographic patterning is insensitive to the number, complexity and layout of oligonucleotides. Oligonucleotide length is essentially unrestricted, although synthesis time is approximately proportional to sequence length. In the case of the image in Fig. 2 (up to 85mers), the synthesis time was ~3 hours.”

5. In level-two painting, the authors further introduced UV exposure to induce crosslinking of oligos. But the UV exposure has been introduced for activating DNA synthesis in level-one painting. Is there any interference about the repeated use of UV light?

Authors reply: There is no interference because once the first level of patterning is complete the photolabile groups used in this step are gone, leaving only DNA tethered to the surface. To clarify this point we have added the following on page 5:

“Level-Two crosslinking reactions can use the same UV light source but does not interfere with Level-One patterning since crosslinking proceeds only after synthesis on the biochip is complete and no photoactive groups remain on the surface.”

6. Why does the crosslinking efficiency decrease as cycles accumulated in figure 3d?

Authors reply: This pattern is only evident in the second graph of 3d (free psoralen). We addressed this in the Supplementary Information (page S14), but are uncertain as to the reason.

7. The author claimed “and indeed, this is the case for tethered psoralen and CNVK, but not for free psoralen (Fig. 3d)” on line 158 to 159 in page 6. However, figure 3d doesn’t give any evidence on it. I am also confused why the figure 3c is identical to figure S3.

Authors reply: This sentence is referring to the increases in the graphs in Figure 3d for CNVK and for tethered psoralen, but not for free psoralen. This may be confusing because of the graph labeling issue arising from this reviewer’s comment #3 and we hope that correcting the y-axis labels for these graphs from crosslinking efficiency to crosslinking yield will help. Some of the figures in the SI (Figure S3 and the top panel of S5) are duplicated from the main text to support the context of the additional data provided. We have now indicated this duplication in the SI.

8. Finally, some subfigures (e. g., figure 3a, b) were not described in the main text, and many figures in the SI were not quoted in the main text.

Authors reply: The sublabels a and b in Figure 3 are intended only for use in the figure caption. Also, several figures in the SI are not referred to in the main text as these are intended as purely supplementary and are not necessary to follow the discussion in the main text.

Reviewer #2 (Remarks to the Author): Authors developed a three-level photolithographic patterning approach: first, photolithographic in situ nucleic acid synthesis for maskless fabrication of oligonucleotide microarrays by a digital micromirror array technique, second photo-crosslinking, and third sequence modifications by nucleic acid processing enzymes. A potential 512-color palette on a grid of 256 × 332 pixels was realized for a Picasso’s *Femme assise* painting in the first level, while nucleic acid crosslinking is used for steganography to conceal an image in a complex pattern in the second level. The results shown in the manuscript are of broad interest to the nano-device community for creation of functional biochip by controlling chemical identity, nucleic acid sequence and surface patterns. I recommend publication of the manuscript after substantial revisions.

Authors comment: We thank the reviewer for recommending publication of our manuscript and for the constructive feedback.

1. My major concern is that in the first level, although a painting of Picasso's *Femme assise* is marvelous by a DNA hybridization approach, author should not offer this as an example for validity of their method. Instead, they should give a details for the principle of their palettes. How in principle to realize the different colors? and how in experiment to realize the component or ratio of RED, GREEN and BLUE? They should compare the colors they obtained to the standard colors by selected colors, even though not totally 512 colors.

Authors reply: Since reviewer #1 also requested more details on the color generation we have elaborated the description of the process in the paragraph above figure 2:

"To demonstrate the very high level of multiplexing versatility of Level-One patterning, we used in situ DNA synthesis to reproduce, at high resolution and at a highly miniaturized scale, a well-known painting by Picasso. To generate a color image, we separated the original image into red, green and blue channels and assigned each grey level within each channel to DNA sequences fully or partially complementary to Cy5 (red), Cy3 (green) or fluorescein (blue) labeled oligonucleotides. For simplicity, only 8 shades of grey (intensity levels) per channel were used, resulting in a red-green-blue color model (RGB) reproduction with 512 unique colors encoded on a grid of 256 × 332 pixels, each with one of 512 surface-bound oligonucleotides (Fig. 2). To generate the greyscale, we chose 8 oligonucleotides per channel with different melting temperatures that were calibrated to result in 8 uniformly spaced fluorescence intensities when hybridized with the common labeled complement. We used sequence truncations rather than mismatches to reduce melting temperatures, as this approach is simpler and reduces biochip synthesis time. The depth of the color space can be increased beyond 512 by also using mismatches to generate finer gradations of melting temperatures. Accurate color rendition requires a calibration curve generated experimentally on a biochip since fluorescence intensity also depends strongly on the specific nucleobase context³². This first level of photolithographic patterning is insensitive to the number, complexity and layout of oligonucleotides. Oligonucleotide length is essentially unrestricted, although synthesis time is approximately proportional to sequence length. In the case of the image in Fig. 2 (up to 85mers), the synthesis time was ~3 hours."

Extensive additional details on the generation of the images are provided in the SI.

2. 42 °C was chosen for hybridization temperature in manufacturing microarray, why? What happen if at other temperatures?

Authors reply: There is nothing special about the 42 °C, nor the details of the hybridization conditions. It is just one of many interchangeable protocols that were developed originally for gene expression microarray analysis. Any hybridization protocol for oligonucleotides in a length range between 25 mers and 80mers and with GC content of about 50% would work similarly. To refer the reader to a literature starting point on microarray hybridization, we have added the following reference to the methods section: 10.1093/nar/28.22.4552

3. In the third level, authors should provide an example for application, as that of image steganography in second level.

Authors reply: We did include an example of an application of Level-Three patterning as shown in Figure 4 and described on page 7. In particular we used a DNA polymerase on a branched DNA structure to create an array of double-stranded DNA with both strands attached to the surface via a common linker. The template strand was enzymatically degradable via uracil DNA polymerase. Due to time and budget constraints we were unable to include a more artistic image demonstration as in Figures 2 and 3.

Reviewers' Comments:

Reviewer #1:

Remarks to the Author:

I think the authors have satisfactorily responded to most concerns. This is a very good paper that merits publication in Nature Communications.

Reviewer #2:

Remarks to the Author:

Providing detailed steps to achieve color is commendable, but I am concerned that it should be necessary to display the palette and compare it to the standard color. Authors should choose several colors to provide experimental details on how to achieve these colors. For example, how to realize RGB (200, 100, 150) and evaluate its deviation from standard RGB (200, 100, 150), and try to discuss or suggest how to improve the image quality in future works. In fact, the color in the rightmost image in Figure 2 is different from the color in the leftmost image, which should be attributed to the deviation of the palette from the standard color.

Reviewers' comments:

Reviewer #1 (Remarks to the Author): I think the authors have satisfactorily responded to most concerns. This is a very good paper that merits publication in Nature Communications.

Authors comment: We thank the reviewer for reading the revised manuscript and for the kind remark.

Reviewer #2 (Remarks to the Author): Providing detailed steps to achieve color is commendable, but I am concerned that it should be necessary to display the palette and compare it to the standard color. Authors should choose several colors to provide experimental details on how to achieve these colors. For example, how to realize RGB (200, 100, 150) and evaluate its deviation from standard RGB (200, 100, 150), and try to discuss or suggest how to improve the image quality in future works. In fact, the color in the rightmost image in Figure 2 is different from the color in the leftmost image, which should be attributed to the deviation of the palette from the standard color.

Authors comment: We thank the reviewer for the additional reading of the manuscript and for the additional remark. We apologize for possibly misunderstanding the request for additional details about the color palette in the previous revision. We originally interpreted this comment to be related to the generation of the three colors (RGB), rather than how they are combined to generate the full color palette. To address this comment we have added additional detail on color rendering accuracy and limitations as well as possible approaches to improve color accuracy and to extend the color palette beyond 3 bits per channel. These additional details are in a new section of the Supporting Information with the heading **“1.h Palette and color rendering accuracy”**

Reviewers' Comments:

Reviewer #2:

Remarks to the Author:

Authors have properly addressed this issue and the paper is publishable with the present version.